# Prediction-based highly sensitive CRISPR off-target validation using target-specific DNA enrichment

Seung-Hun Kang [1,2], Wi-jae Lee[2,3], Ju-Hyun An[2,4], Jong-Hee Lee[5], Young-Hyun Kim[4,5], Hanseop Kim[2,6], Yeounsun Oh[2,7], Young-Ho Park[2], Yeung Bae Jin[5], Bong-Hyun Jun [3], Junho K. Hur [8,9,10✉], Sun-Uk Kim[2,4✉] & Seung Hwan Lee [5✉]

CRISPR effectors, which comprise a CRISPR-Cas protein and a guide (g)RNA derived from the bacterial immune system, are widely used for target-specific genome editing. When the gRNA recognizes genomic loci with sequences that are similar to the target, deleterious mutations can occur. Off-target mutations with a frequency below 0.5% remain mostly undetected by current genome-wide off-target detection techniques. Here we report a method to effectively detect extremely small amounts of mutated DNA based on predicted off-target-specific amplification. In this study, we used various genome editors to induce intracellular genome mutations, and the CRISPR amplification method detected off-target mutations at a significantly higher rate (1.6~984 fold increase) than an existing targeted amplicon sequencing method. In the near future, CRISPR amplification in combination with genome-wide off-target detection methods will allow detection of genome editor-induced off-target mutations with high sensitivity and in a non-biased manner.

[1] Department of Medicine, Graduate School, Kyung Hee University, Seoul, Republic of Korea. [2] Futuristic Animal Resource & Research Center (FARRC), Korea Research Institute of Bioscience and Biotechnology (KRIBB), Cheongju, Republic of Korea. [3] Department of Bioscience and Biotechnology, Konkuk University, Seoul 143-701, Korea. [4] Department of Functional Genomics, KRIBB School of Bioscience, Korea University of Science and Technology (UST), Daejeon, Korea. [5] National Primate Research Center (NPRC), Korea Research Institute of Bioscience and Biotechnology (KRIBB), Cheongju, Korea. [6] School of Life Sciences and Biotechnology, BK21 Plus KNU Creative BioResearch Group, Kyungpook National University, Daegu, Republic of Korea. [7] Division of Biotechnology, College of Life Science and Biotechnology, Korea University, Seoul 02841, Republic of Korea. [8] Department of Pathology, College of Medicine, Kyung Hee University, Seoul, Republic of Korea. [9] Department of Biomedical Science, Graduate School, Kyung Hee University, Seoul, Republic of Korea. [10] Department of Medical Genetics, College of Medicine, Hanyang University, Seoul, Republic of Korea. ✉email: jh0210@gmail.com; sunuk@kribb.re.kr; lsh080390@kribb.re.kr

The CRISPR–Cas system, consisting of various Cas proteins and guide (g)RNA, is a bacterial or archaea immune system required to defend viral DNA through target sequence specific cleavage. The CRISPR–Cas system allows targeted editing of genes of interest in various organisms, from bacteria to humans, and has versatile applications in vivo[1–3]. However, when the CRISPR system recognizes sequences similar to the target sequence, deleterious off-target mutations can occur, especially in mammals given their large genomes, which can lead to malfunctions[4]. Off-targeting issues due to guide (g)RNA characteristics have been reported for various CRISPR effectors, such as Cas9 and Cas12a (Cpf1)[3,5–8]. Consistently, there is a need for precise control of CRISPR–Cas function, and in particular, for a method to develop CRISPR effectors[9–11] that precisely target a desired locus while avoiding off-target effects, before CRISPR effectors can be introduced for human therapeutic purposes[12].

To date, various methods, mostly based on next-generation sequencing (NGS), have been developed to detect the off-target mutations in vivo[5,13–15]. These methods provide genome-wide detection of off-target mutations in an unbiased fashion both, inside and outside the cell. However, the methods often result in ambiguous or no detection of off-target mutations, particularly, when they are below the sequencing error rate (<0.5%)[16]. In addition, these methods detect non-common variations other than shared off-target mutations, and thus require additional validation. In order to measure a small amount of off-target mutations caused by the above effectors with high sensitivity, a method to enrich and relatively amplify the mutated over the much more abundant wild-type DNA using CRISPR endonucleases based on previous method (CRISPR-mediated, ultra-sensitive detection of target DNA (CUT)-PCR) was developed[17]. First, all off-target candidate sequences similar to the given CRISPR effector target sequence are selected by in silico prediction[18]. Subsequently, the wild-type DNA that does not contain mutations in each of the off-target candidate sites is eliminated by the effector to enrich mutant DNA, which is then PCR-amplified, thus enabling the detection of extremely small amounts of mutant DNA with a sensitivity superior to that of conventional deep sequencing methods. CRISPR amplification based off-target mutation enrichment was never tried with genomic DNA before and we further enhanced this method with direct amplification on cell-extracted genomic DNA.

With the rapid advancements in CRISPR–Cas genome-editing technologies, their application in human gene therapy is being considered[1,19]. Therefore, there is an urgent need for a method to accurately detect whether or not CRISPR effectors operate at unwanted off-target sites. In this study, we develop a method for detecting very small amounts of off-target mutations (below the detection limit of conventional amplicon sequencing) derived from genetic modification of specific sequences using CRISPR-amplification technology in vivo with high accuracy.

## Results

**Mutant DNA fragment enrichment with CRISPR amplification**. The existing NGS methods have limited sensitivity in detecting genome-wide off-target mutations induced by CRISPR effectors in vivo. To overcome this limitation, we developed a method for amplifying and analyzing a small amount of mutations on CRISPR effector treated genomic DNA. To detect the off-target mutations, we designed a CRISPR amplification method to allow relative amplification of a small amount of mutant DNA fragments over wild-type DNA. The principle of this method is shown in (Fig. 1a). First, the off-target sequences similar to the target sequence are predicted in silico. Then, genomic DNA is extracted from CRISPR-edited cells, and on-

target and predicted off-target genome sequences are PCR-amplified. Subsequently, the amplicons are processed by the CRISPR effector (using the same or optimally designed gRNA to remove wild-type DNA other than mutant DNA), and the enriched mutant DNA fragments are PCR-amplified. Amplicons obtained by three times repeated CRISPR effector cutting and PCR amplification are then barcoded by nested PCR and subjected to NGS. The indel frequency (%) is calculated to evaluate the presence of off-target mutations. To verify that mutated DNA was indeed relatively amplified over non-mutated DNA by this method, we edited HEK293FT cells using the CRISPR–Cas12a system to induce indels in target sequences and applied the CRISPR amplification method to genomic DNA samples extracted from the cells (Fig. 1b–d). As shown in (Fig. 1b), targeted amplicon sequencing and genotyping (Supplementary Fig. 1a) enabled the detection of extremely low concentrations of mutant DNA fragments (~1/100,000%) through multiple rounds of amplification by wild-type DNA specific cleavage (Fig. 1c, Top). Only one cycle of CRISPR amplification was used to detect mutant genes with the rate around 0.1%, and three repeated assays allowed determining the indel mutation rate as low as 0.00001% (Fig. 1c, Bottom). When analyzing the pattern of mutations that were relatively amplified by CRISPR amplification up to three times for each diluted sample from original CRISPR effector treated genomic DNA, deletion patterns were observed mainly in the CRISPR–Cas12a target sequence, in line with previous reports[20,21] (Fig. 1d). When the wild-type DNA was removed by the CRISPR effector, DNA fragments containing deletion-type mutations (Fig. 1d, right) of various sizes were effectively amplified over non-mutated DNA fragments. The deletion pattern for the CRISPR–Cas12a effector revealed that DNA fragments with large deletions tended to be more effectively amplified than fragments with smaller deletions (Fig. 1d, left; Supplementary Fig. 9). The results indicated that a 1-bp alteration from the original sequence within the protospacer region leads to a high probability of re-cleavage by the CRISPR effector, and there was a general tendency for more efficient amplification of large deletions, with some exceptions of preferred sequences (Supplementary Figs. 4a–d, 5, and 6a).

**Off-target cleavage of CRISPR–Cas12a effector**. Previous studies showed that CRISPR–Cas12a targets thymine-rich regions and induces less off-target mutations than CRISPR–Cas9 because it is less tolerant for mismatches[3,7]. Nonetheless, off-target mutations with rates below the NGS sensitivity limit (<0.5%) may occur, which can potentially cause serious problems, particularly in therapeutic settings. To address the safety issue, we assessed whether CRISPR amplification allows the detection of a small amount of Cas12a-induced off-target mutations with high sensitivity (Fig. 2). A targeted sequence (*RPL32P3* site) in U2OS (Fig. 2a; Supplementary Fig. 1b) and HEK293FT cells (Supplementary Fig. 2) was analyzed for Cas12a-induced mutations measured as indel frequencies (0.1–11.2%) in off-target candidate loci predicted by in silico analysis[18] (Supplementary Table 2). Ten predicted off-target sequences were processed using an optimally designed Cas12a–crRNA complex (ideally designed to cleave wild-type DNA for each target and off-target sites, respectively (Supplementary Fig. 13a)) to enrich the small amount of mutated DNA. The results indicated that mutant DNA amplification occurred at various indel frequency (99.9, 99.7, 97.5, 98.6 and 65.7% for off-target1-5, Fig. 2a), depending on the amount of off-target mutant DNA. The indel frequency for the on-target *RPL32P3* site (Supplementary Table 2) in U2OS cell was 7.7% (Fig. 2a). After three rounds of CRISPR amplification, the mutant DNA fragment was amplified with an efficiency close to 100%

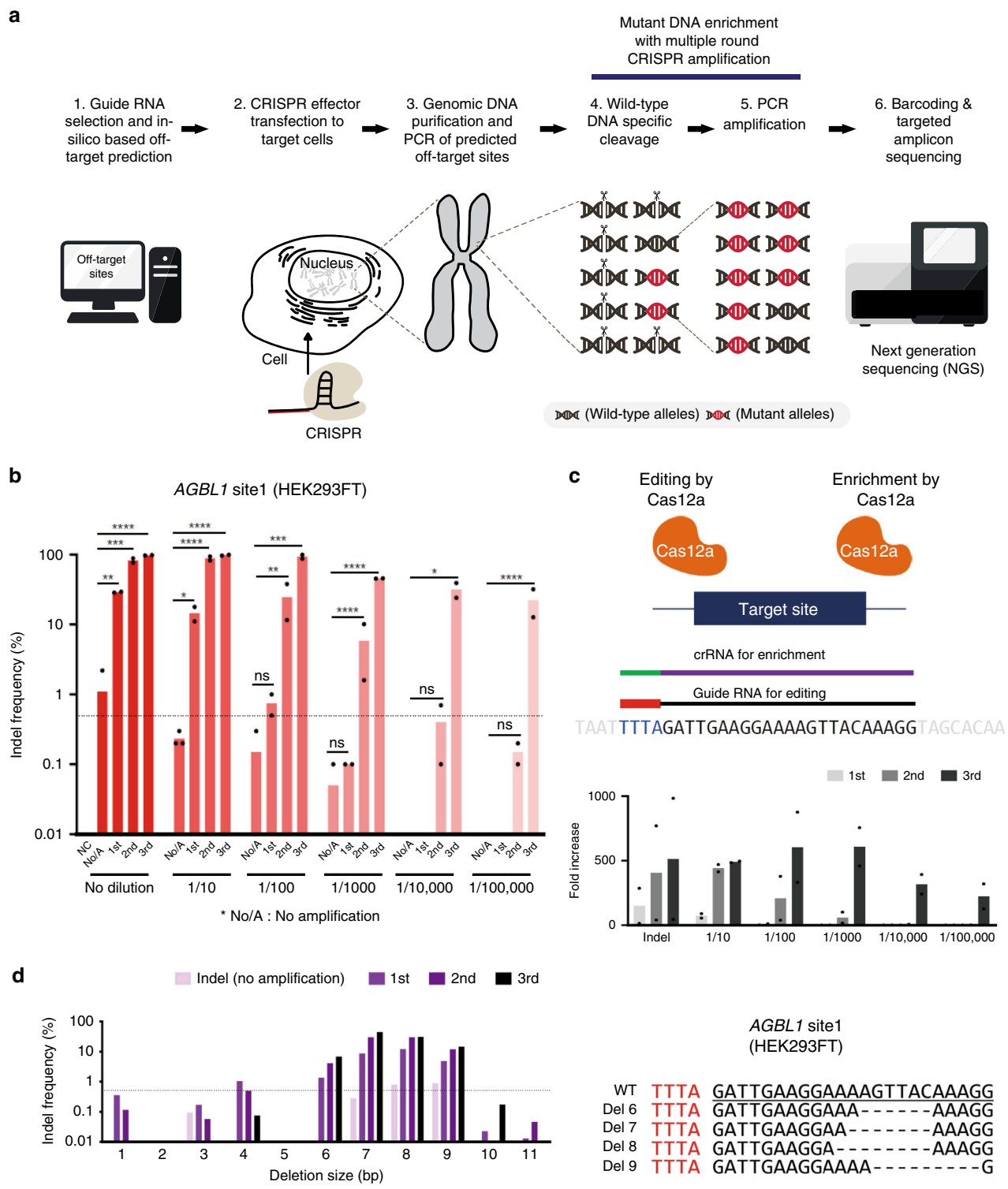

(12.8-fold enriched). In a negative control treatment without Cas12a, the indel frequency could not be determined (~0%) by three rounds of CRISPR amplifications. When off-target mutation detection by CRISPR amplification was compared with the conventional targeted amplicon sequencing[22] or the result of GUIDE-seq technology[3], candidate off-target sequences 1–5 (Supplementary Table 2) were commonly detected by CRISPR amplification (26.1, 41.2, 37, 155.5, and 117-fold enriched for off-target1-5), and the remaining candidate off-target sequences 6–10 (Supplementary Table 2) did not show mutations (Fig. 2b; Supplementary Table 4). In particular, CRISPR amplification was able to accurately identify off-target mutations (off-target loci 4 (HEK293FT) and 5(U2OS)) that were difficult to identify by conventional targeted amplicon sequencing due to the low copy number of mutant DNA (Fig. 2a; Supplementary Fig. 2). The Cas12a-induced indel pattern revealed that off-target mutations were mainly deletions rather than insertions (Supplementary Fig. 4). In particular, amplification of insertions was not observed above 3rd-order enrichment. Overall, DNA fragments with large deletions around the cleavage site tended to be more effectively

**Fig. 1 Schematics of the detection of off-target mutations using the CRISPR amplification method. a** Workflow of the off-target detection method using CRISPR amplification. (1) Based on candidate off-target sequences predicted in silico, a gRNA is designed to enrich off-target sequences by subsequent CRISPR cleavage. (2) The CRISPR effector is transfected into cells to induce specific mutations. (3) Genomic DNA is extracted from the cells and predicted off-target loci are PCR-amplified. (4) The PCR amplicons are cleaved by the CRISPR effector. (5) DNA with non-cleaved indel mutations are amplified preferentially over wild-type DNA. (6) The amplified DNA is barcoded and analyzed by NGS. **b** Quantitative analysis of mutant DNA enrichment by CRISPR amplification. Genomic DNA samples with mutations induced in the target sequence were serially diluted (up to 1/100,000) and amplified. The indel efficiency (%) was calculated by sequencing DNA amplicons obtained from genomic DNA extracted from CRISPR–Cas12a edited cells. The $X$ axis represents the degree of dilution of the genomic DNA samples, and the $Y$ axis represents the indel detection rate (%). NC indicates the negative control of no genome editing. No/A indicates the genome-edited sample, but no amplification. The dashed line indicates the NGS detection limit (=0.5%). Data are shown as mean from two ($N=2$) independent experiments. $P$-values are calculated using a one-way ANOVA, Tukey's test (ns: not significant, $*P=0.0332$, $**P=0.0021$, $***P=0.0002$, $****P=0.0001$). **c** Top: design of the crRNA for editing and mutant DNA enrichment with Cas12a, respectively. crRNA was used to induce target genomic locus mutation by AsCas12a effector and exactly same crRNA was used for mutant DNA enrichment by CRISPR amplification. Bottom: fold increase of indel frequency by CRISPR amplification ($N=2$) for each diluted samples from originally mutation-induced genomic DNA. **d** NGS analysis of AsCas12a-induced indel patterns enriched by CRISPR amplification. Left: gradual amplification (no amplification, primary, secondary, and tertiary amplification) of mutant DNAs with different deletion sizes. Each amplification stage is indicated by different colors. Right: representative enriched mutation patterns from third-round of CRISPR amplification with on-target amplicon. Source data is in the Source Data file.

amplified than DNAs with smaller deletions. We next conducted CRIPSR amplification analyses for CRISPR–Cas12a off-target analysis at several loci after introducing mutations at the *DNMT1* gene in HEK293FT cells (Supplementary Fig. 5). In addition to the indel frequency at the target sequence, the indel frequencies at two out of the three off-target genes predicted in silico (Supplementary Table 2) were determined (Supplementary Figs. 1c and 5a, b). Obviously, indel mutations in off-target locus 2 that were difficult to detect by targeted amplicon sequencing were identified by CRISPR amplification (Supplementary Fig. 5 and Supplementary Table 5), and in off-target locus 1, no significant mutation was detected by this method. The indel pattern for each locus analyzed by sequencing revealed that deletions rather than insertions were the major enriched mutations in target and off-target sequences in the *DNMT1*-site3 locus (Supplementary Fig. 5c–e).

**Off-target mutation detection for the CRISPR–Cas9 effector**. To verify the potential of the Cas9 effector for inducing off-target mutations at the cellular level, CRISPR amplification was applied to detect candidate off-target mutations (Supplementary Table 2) predicted based on the target sequence in HEK293FT edited with CRISPR–Cas9 (Fig. 3). In order to selectively amplify the mutant allele via specific cleavage of wild-type DNA using Cas12a, the seed region of CRISPR–Cas9 target sequence (*FAT3* site1) was designed to harbor the PAM sequence (TTTN) of Cas12a (Supplementary Fig. 13c). CRISPR amplification revealed a significant increase in indels in the target DNA, which was confirmed by NGS (Fig. 3a) and DNA cleavage assays (Supplementary Fig. 3a). The amplification of on-target intracellular indels induced by CRISPR–Cas9 increased (34.4 fold) with consecutive rounds of CRISPR amplification (Fig. 3b). The indel pattern showed a gradual increase in >2-bp deletions caused by CRISPR–Cas9 (Supplementary Fig. 6a). A small amount of insertion mutations of various sizes was detected, but no significant amplification was detected. CRISPR amplification was also performed for the predicted off-target loci (Supplementary Table 2), and unique indel pattern was detected at off-target site 3 (Fig. 3a, b; Supplementary Fig. 6b). Conventional NGS did not allow detection of the off-target indels with frequencies below the detection limit (indel frequency ~ 0.5)[23–27]. However, when the amplicon was subjected to third rounds of CRISPR amplification, even 1-bp-deletions were detected at a significant rate (51-fold increase vs. without amplification) (Fig. 3b; Supplementary Fig. 6b and Supplementary Table 6). Next, to assess whether the enrichment system can be universally applied to accurately detect the off-target mutations caused by CRISPR–Cas9, we applied the CRISPR

amplification method to additional target sites. We revisited a previous study that conducted Guide-seq and showed that genome editing of *HEK 293 site4* and *RNF2* locus shows extremely low amount of off-target mutations, indicated by Guide-seq read numbers (Supplementary Tables 7 and 8)[5]. We performed CRISPR amplification for off-target sites where the Guide-seq read count was extremely low (off-target site 4, 5, 6 of *HEK 293 site4*) and it was difficult to verify the authenticity (Fig.3c; Supplementary Fig. 7). The results showed that significant amplification was detected for off-target5 (*HEK 293 site4*), whereas no significant amplification was detected for off-target4, 6 (*HEK 293 site4*), respectively (Fig. 3c, d). For the endogenous off-target sites of the *RNF2* target sequence, which show 0 Guide-seq reads, there was no significant detection through CRISPR enrichment (Supplementary Fig. 8 and Supplementary Table 8). The results indicated that the current genome-wide off-target detection methods are prone to identifying false positive off-targets with low abundances, and the CRISPR amplification enables sensitive screening of false positives and allows verifying the true-positive off-targets.

We further investigated whether our system can robustly enrich small DNA mutations such as ±1 indels generated by CRISPR–Cas9 (Supplementary Figs. 9, 10 and 11). First, in order to confirm that the 1 bp mutation can be effectively amplified, mutant DNA amplicons containing 1 and 10 bp deletion on the same target site, respectively, were separately prepared by PCR (Supplementary Fig. 9a). Next, we compared the CRISPR amplification that enriched the 1 and 10 bp deletion mutant DNA that were mixed with wild-type DNA in same amounts. The result showed that the DNA amplicon containing 1 bp deletion could be efficiently amplified (Supplementary Fig. 9b, left). However, the DNA containing 1 bp deletion is relatively less amplified (14.1 (−1 Del only)/1.5 (mixed) fold enriched) when compared to the DNA containing 10 bp deletion (19.3 (−10 Del only)/41.9 (mixed) fold enriched) in individual (Supplementary Fig. 9b, c) or mixed amplification conditions (Supplementary Fig. 9d–f). After the artificial DNA amplification tests, we next applied CRISPR amplification to the endogenous sites (Spacer 8A, Spacer 15A) of genomic DNA from the HEK293FT cells where 1 bp indels were shown to be predominant in CRISPR–Cas9-induced mutations, in a previous study[28]. To this end, we delivered CRISPR–Cas9 to HEK293FT cells, confirmed that 1 bp indels were mainly formed in the same locus (Supplementary Figs. 10a and 11a). We then also confirmed that mutations were effectively enriched by CRISPR amplification (Supplementary Figs. 10b–e and 11b–d). In the Spacer 8A site, the +1 insertion pattern was predominant in the mutation repertoire, and in the

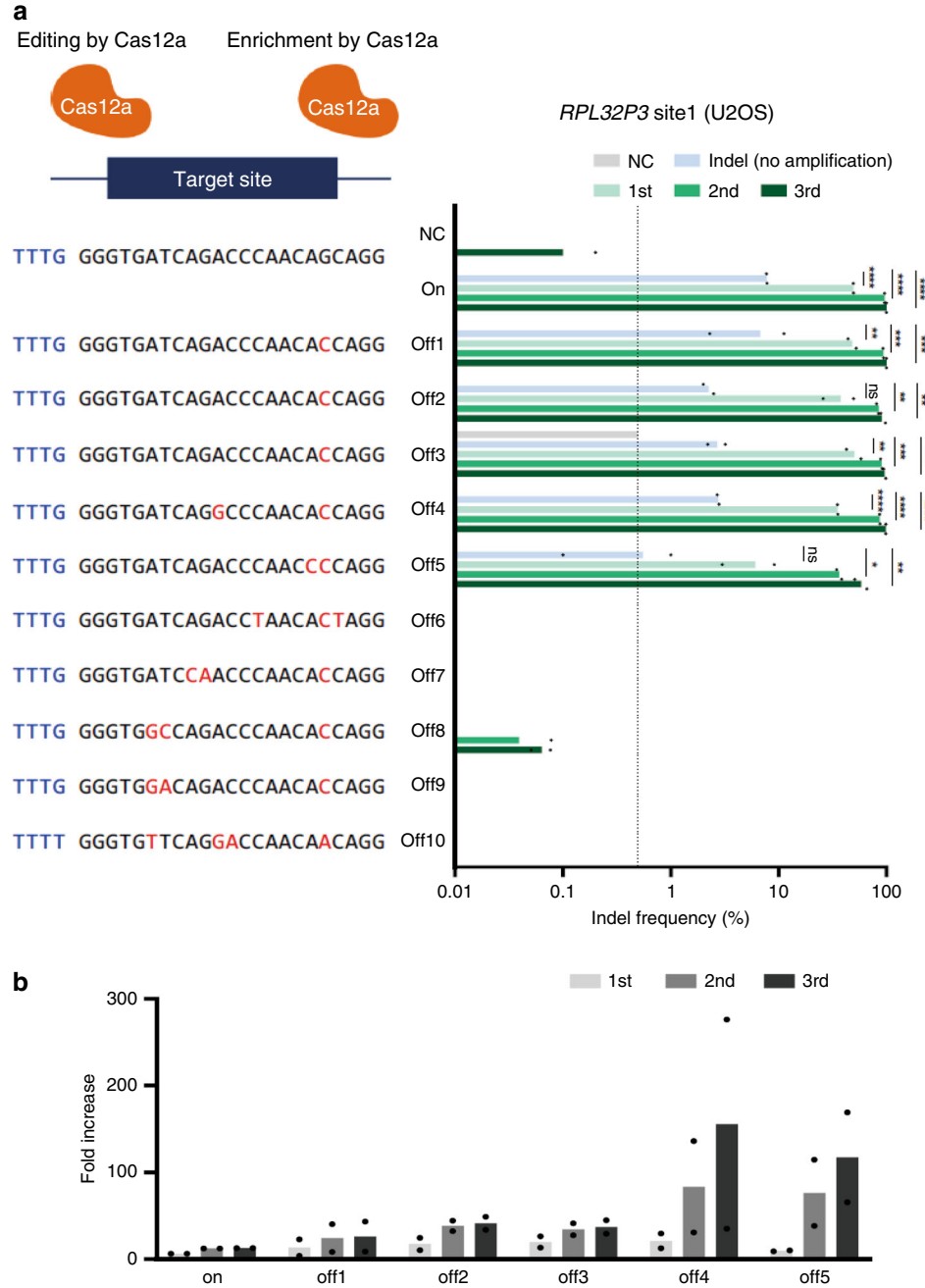

**Fig. 2 Detection of off-target mutations induced by CRISPR–Cas12a. a** Detection of off-target mutations for the target sequence (*RPL32P3* site) generated by the CRISPR–Cas12a effector in U2OS cells. PCR amplicons were generated for 10 off-target sequences[3] similar to the target sequence and the indel frequency (%) was determined by NGS after sequential amplifications with CRISPR–Cas12a. Each round of CRISPR amplifications are marked by different colors (light blue: no amplification, blue: 1st CRISPR amplification, green: 2nd CRISPR amplification, dark green: 3rd CRISPR amplification). NC indicates a negative control for no Cas12a delivery into the cells. The *Y* axis represents the amplified target and off-target sequences, and the *X* axis represents the indel frequency (%) in the analyzed amplicon on a log scale. The dashed line indicates the NGS detection limit (=0.5%). Data are shown as mean from two ($N = 2$) independent experiments. *P*-values are calculated using a one-way ANOVA, Tukey's test (ns: not significant, *$P = 0.0332$, **$P = 0.0021$, ***$P = 0.0002$, ****$P = 0.0001$). **b** Fold increase after CRISPR amplification for each target and off-target sequence. Primary, secondary, and tertiary CRISPR amplification ($N = 2$) results are shown in gray, dark gray, and black, respectively. Source data is in the Source Data file.

process of amplifying it, the +1 insertion pattern was effectively enriched (Supplementary Fig. 10d, e). In particular, a −10 bp deletion pattern that was initially below the NGS detection level, was significantly enriched (Supplementary Fig. 10e). Similarly, in the Spacer 15A site, mutations in the form of +1 insertion and −1 deletion were mainly formed simultaneously (Supplementary

Fig. 11a, d, top), but notably, in the process of amplification, various deletion types of mutations that were below the NGS detection level were amplified more effectively (Supplementary Fig. 11d).

We next sought to compare the CRISPR amplification between wild-type SpCas9 (wt-SpCas9) and eSpCas9, an engineered form

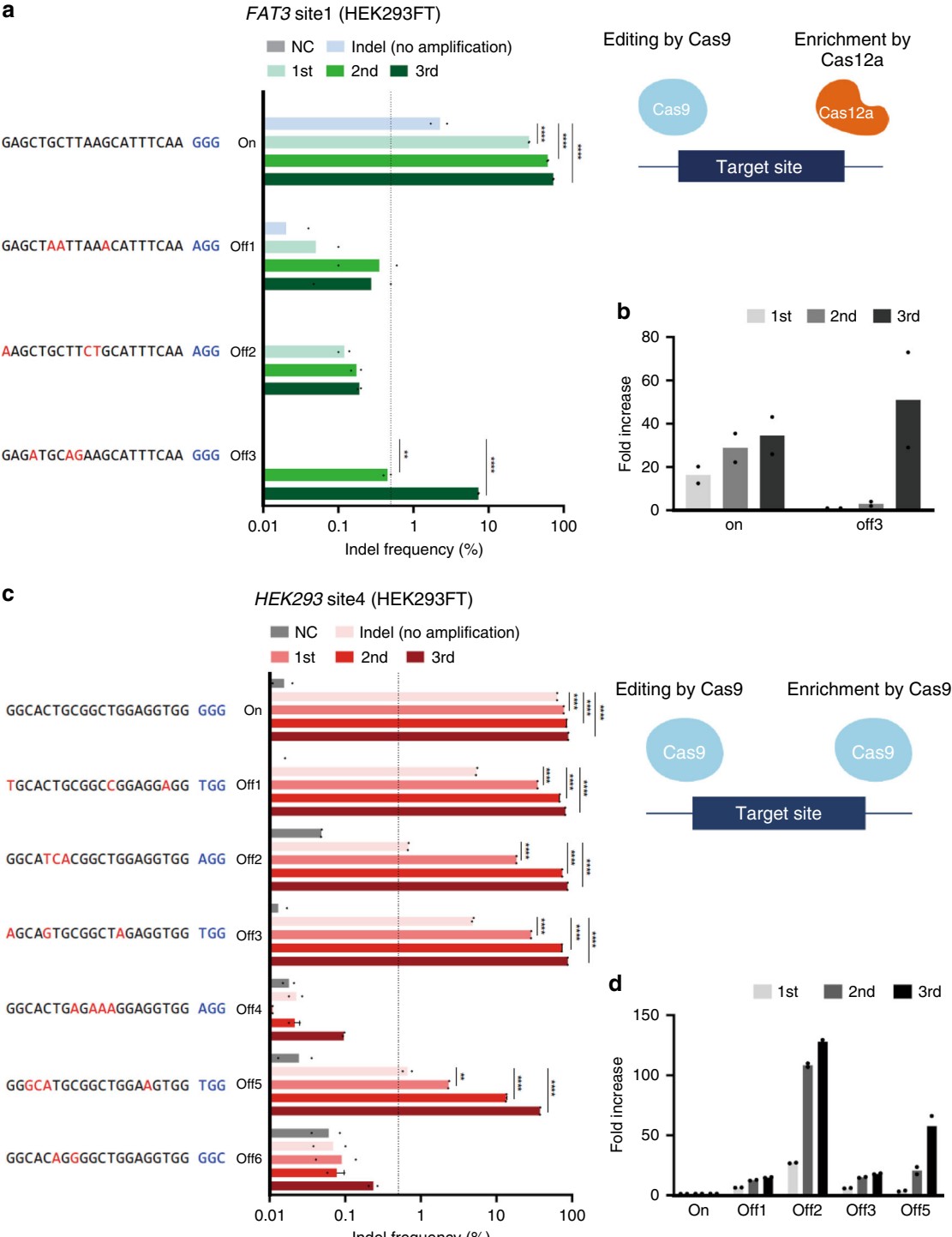

of SpCas9 with higher specificity[11]. To this end, we applied CRIPSR amplification using purified wt-SpCas9 and eSpCas9 to mutations formed by genome editing at the Spacer 15 A site (Supplementary Fig. 11d, middle, bottom, 11e). Overall, the mutation analyses showed that the −8, −10, and −42 bp deletion patterns were preferentially amplified than 1 bp size of indel forms. However, the amplification patterns were distinct between wt-SpCas9 and eSpCas9 (Supplementary Fig. 11e). In the first and second round of amplification, wt-SpCas9 showed relatively low amplification efficiency of 1-bp indel compared to larger indels. Notably, on the other hand, amplification by eSpCas9 resulted in enrichment of −1 bp mutation patterns. These data indicated that

eSpCas9 showed enhanced ability for amplifying small size indels in the process of removing background wild-type DNA.

**Single-base substitution (detection for adenine base editor)**. In addition to the insertions/deletions caused by target DNA cleavage by the CRISPR effector, we evaluated whether the CRISPR amplification can be used to detect single-base changes caused by off-target effects of adenine base editor (ABE)[29] (Fig. 4). To this end, we first transfected, ABE and gRNA (Supplementary Table 1) expression vectors (ABEmax version, addgene no. 112098) into HEK293FT cells to induce single-base substitutions

**Fig. 3 Detection of off-target mutations induced by CRISPR–Cas9. a** Left: detection of off-target mutations for the target sequence (*FAT3* site1) generated by the CRISPR–Cas9 effector in HEK293FT cells. PCR amplicons were generated for on-target and three off-target sequences predicted in silico and the indel frequency (%) was analyzed by NGS after sequential CRISPR amplifications. The *Y* axis represents the amplified target and off-target sequences, and the *X* axis represents the indel frequency (%) in the analyzed amplicon on a log scale. NC indicates a negative control for no CRISPR–Cas9 delivery into the cells. Each amplification stage for mutant DNA enrichment is shown in gray (NC), light blue (no amplification), cyon (1st CRISPR amplification), green (2nd CRISPR amplification), and dark green (3rd CRISPR amplification). The dashed line indicates the NGS detection limit (=0.5%). All experiments were conducted at least two times. Data are shown as mean from two (*N* = 2) independent experiments. *P*-values are calculated using a one-way ANOVA, Tukey's test (ns: not significant, \**P* = 0.0332, \*\**P* = 0.0021, \*\*\**P* = 0.0002, \*\*\*\**P* = 0.0001). Right: schematics of the target-specific editing with Cas9 and mutant DNA enrichment with Cas12a, respectively. **b** Fold increases in *FAT3* target and off-target mutant DNA after CRISPR amplification (*N* = 2). **c** Left: detection of off-target mutations for the target sequence (*HEK293* site4) generated by the CRISPR–Cas9 effector in HEK293FT cells. PCR amplicons were generated for on-target and six off-target sequences according to the previous study[5] and the indel frequency (%) was analyzed by NGS after sequential CRISPR amplifications. Each amplification stage for mutant DNA enrichment is shown in gray (NC), light pink (no amplification), dark pink (1st CRISPR amplification), red (2nd CRISPR amplification), and dark red (3rd CRISPR amplification). All experiments were conducted at least two times. Data are shown as mean from two (*N* = 2) independent experiments. *P*-values are calculated as in **a**. Right: schematics of the target-specific editing with Cas9 and mutant DNA enrichment with Cas9, respectively. **d** Fold increases in target (*HEK293* site4) and off-target mutant DNA after CRISPR amplification (*N* = 2). In **b** and **d**, primary, secondary, and tertiary CRISPR amplification fold increase are shown in light gray, dark gray, and black. Source data is in the Source Data file.

at on-target and potential off-target genomic loci (Supplementary Table 2). Next, the genomic DNA extracted from the cells was subjected to CRISPR amplification for candidate off-target sequences predicted based on the gRNA sequence for the target DNA (*PSMB2*) (Fig. 4a). To specifically amplify base-changed over wild-type DNA, the PAM sequence (TTTN) recognized by Cas12a was placed in a window where the base substitution is mainly generated in the target sequence (Fig. 4a, right; Supplementary Fig. 13d). If the PAM sequence recognized by Cas12a has a base substitution, wild-type DNA without base substitution can be removed. Three cycles of CRISPR amplification on samples treated with ABE (*PSMB2* site) (Supplementary Table 1) revealed that DNA fragments with a base substitution (A>G) in the ABE editing window were significantly amplified (75.6% substitution, 7.67 fold increase) (Fig. 4b). Consistent with the previously reported properties of adenine base editor[29], The base substitution (A>G) was found to occur mainly at adenine sites in the 3–9 bp window within the ABE target sequence (Fig. 4c). The detected frequencies of the base substitution gradually increased with increasing cycles of CRISPR amplification. Next, CRISPR amplification was applied to predicted off-target sites (Supplementary Table 2). The A>G base substitution was significantly induced in each of the off-target sequences (off-targets 2, 3, and 4 in Fig. 4a, c). A low amount of base substitution could not be detected by NGS, but it was detected by CRISPR amplification (434, 272.5, and 58 fold increase, respectively) (Fig. 4b; Supplementary Table 9). Interestingly, in contrast to on-target sequences, where the A>G base substitution occurred evenly within the window of the guide sequence (corresponding to 3–9 nt positions of sgRNA from the PAM distal end), the three off-target sites showed intensive base substitutions at the adenines at the 3rd position from the PAM sequence (Fig. 4c).

## Discussion
As the CRISPR system is based on gRNAs that bind complementarily to target genes, it often causes mutations at unintended genomic loci with similar sequences. Efforts have been made to identify the presence of the off-target mutations, and to prevent their formation. In order to overcome the limitations regarding the sensitivity of current methods based on genome-wide analysis, we developed a method to amplify and detect small amounts of off-target mutations in intracellular genes generated by CRISPR effectors. The key of the mutated gene amplification method using CRISPR, developed in this study, is to enrich mutant genes containing off-target sequences predicted in silico and to remove the background wild-type DNA before PCR

amplification. This selective and iterative enrichment method resulted in exponential increment of sensitivity to detect off-targets. On the other hand, previous off-target detection methods detected the in vitro CRISPR cleavage events of the off-targets per se[14,15]. In such schemes, iterative DNA digestion can only provide linear, but not exponential, increase in the numbers of off-target reads in deep sequencing data sets.

As the CRISPR effectors, currently in use, sensitively recognize PAM sequences in the target DNA and the seed sequence of the protospacer, we considered these sequences when designing the gRNA. In particular, in the case of base editors that induce single-nucleotide substitutions, fragments containing mutations can be efficiently amplified by constructing a gRNA such that the PAM sequence is disrupted when a mutation is introduced. Our detection method using CRISPR amplification has various advantages. First, a very small amount of mutant DNA can be enriched in one round of CRISPR amplification, and stepwise amplification shows high sensitivity in detecting mutant DNA fragments of up to (0.00001%). Second, mutant genes can be easily amplified in a high-throughput manner. Third, as a gRNA and CRISPR effector are used, the method can be widely applied to insertions/deletions and single-base substitutions generated by various CRISPR effectors. In addition, it is possible to apply the method to precisely analyze whether mutations are induced in off-target sequences by using new techniques, such as prime editing[30]. One pre-requisite of the current CRISPR amplification method is the need for accurate in silico prediction of potential off-target sites, for which several online tools are available (http://www.rgenome.net/cas-offinder/, http://crispr.bme.gatech.edu, https://portals.broadinstitute.org/gpp/public/analysis-tools/sgrna-design, http://chopchop.cbu.uib.no/, http://www.e-crisp.org/E-CRISP/designcrispr.html, https://rth.dk/resources/crispr/crisproff/)[18,31–36]. Another technical difficulty is that there is a possibility of error incorporation during sequential PCR amplifications. In order to overcome this problem, Phusion High-Fidelity DNA Polymerase[37] with an extremely low error rate was used, and a maximum of three amplification cycles was used based on a negative control experiment, which does not shows significant enrichment of mutation patterns (Supplementary Fig. 12).

In this study, we developed and applied the CRIPSR amplification method to analyze the off-target mutation propensities of CRISPR–Cas12a, CRISPR–Cas9, and ABE. The Cas12a and Cas9 effectors, respectively, recognize T- and G-rich PAM sequences, and bind to the sequences complementary to the gRNA in the target DNA to induce double-helix cleavage. These effectors are

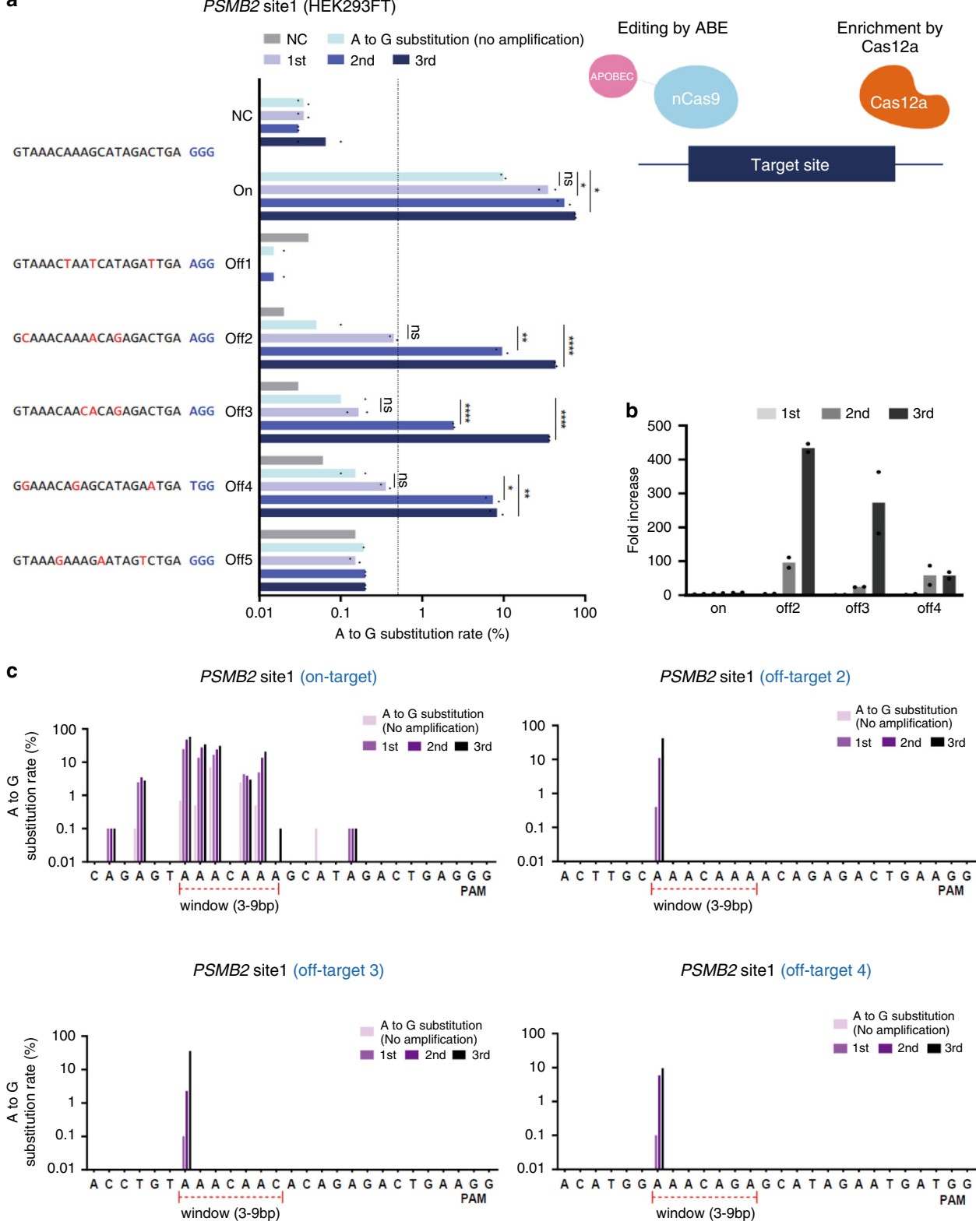

known to be very sensitive to mismatches in the PAM proximal region and less sensitive to mismatches in PAM distal regions. In this study, we compared CRISPR amplification with conventional NGS (next-generation sequencing) or previously reported GUIDE-seq data[3] for the detection of off-target mutations induced by CRISPR–Cas12a. The results of NGS and GUIDE-seq methods identified five potential off-target mutations for

*RPL32P3* site in U2OS or HEK293FT cell, including various mutations in the PAM distal region (Fig. 2a). However, two of the five off-targets could not be confidently confirmed since their mutation frequencies were below the NGS detection levels (0.5%). Our method allowed the amplification and detection of the mutations with low abundance, below the NGS detection level. In addition, three off-target mutations in *DNMT1* were analyzed by

**Fig. 4 Detection of single-base off-target mutations induced by ABE. a** Left: detection of off-target mutations for the target sequence (*PSMB2* site1) generated by ABE in HEK293FT cells. NGS was used to confirm whether single-base mutations in target and non-target sequences can be amplified by CRISPR amplification. The *Y* axis represents the amplified target and off-target sequences, and the *X* axis represents the base substitution (A>G) frequency (%) in the analyzed amplicons on a log scale. NC indicates a negative control for no ABE delivery into the cells. The dashed line indicates the NGS detection limit (=0.5%). Data are shown as mean from two (*N* = 2) independent experiments. *P*-values are calculated using a one-way ANOVA, Tukey's test (ns: not significant, *P = 0.0332, **P = 0.0021, ***P = 0.0002, ****P = 0.0001). Each amplification stage for mutant DNA enrichment is shown in light purple (NC), cyon (no amplification), light blue (1st CRISPR amplification), blue (2nd CRISPR amplification) and dark blue (3rd CRISPR amplification). Right: schematics of the target-specific editing with ABE and mutant DNA enrichment with Cas12a, respectively. **b** Fold increases in single-base substitution rates (%) after CRISPR amplification (*N* = 2) for target and off-target sequences. Primary, secondary, and tertiary CRISPR amplification fold increases are shown in gray, dark gray, and black. **c** Percentages of single-base substitution (A to G) frequency (%) in target and off-target sequences according to no, 1st, 2nd, and 3rd amplification, respectively. The target window (3–9 bp from PAM distal region) is indicated in each panel.

CRISPR amplification (Supplementary Fig. 5), and we observed that off-target indels with a very low read number in conventional NGS were effectively amplified the amplification method. Furthermore, we showed the application of the CRIPSR amplification method to the intracellular on- and off-target mutations induced by CRISPR–Cas9. Similar to the findings for CRISPR–Cas12a, mutations below the NGS detection level were amplified to significance level in multiple in silico predicted Cas9 off-target loci. Notably, the CRISPR amplification efficiently enriched ±1 indel forms in addition to the multiple indel patterns of various sizes generated by CRISPR–Cas9. Interestingly, in addition to the insertion/deletions caused by double-helix cleavage by CRISPR–Cas12a or –Cas9, single-base substitutions at off-target sites induced by ABE were detected at very high significance levels. To cover the broad window of adenine mutation, we considered the presence of PAM sequence to optimally design the guide RNAs for the target sequences. Using adequately designed gRNAs and CRISPR amplification, we found that extremely small amounts of off-target base substitution (A>G) below the NGS detection level could be detected, and the sensitivity was augmented with increasing CRISPR amplification cycles. Depending on the target sequences, it may sometimes be difficult to amplify the various types of mutations with Cas12a, which requires TTN or TTTN sequence for recognition of PAM. Nonetheless, there are methods to alleviate the PAM limitations. First, various CRISPR effectors with diverse PAM sequences could be selectively applied according to target sequences. For example, for the on/off-target DNA enrichment in (Fig. 3c, d), we could apply SpCas9 (NGG recognition as a PAM) to specifically cleave the wild-type DNAs. Second, it is possible to design sgRNAs so that the mutation occurs within the seed sequences proximal to PAM, where CRISPR effectors can discriminate base-pair mismatches with high fidelity. Consistently, CRISPR amplification via designing guide RNAs that perfectly match a wild-type allele could also serve as a method for specific cleavage of wild-type DNA and selective enrichment of mutant alleles. Third, application of engineered SpCas9 with relaxed PAM restriction, such as the recently engineered SpRY[38] or SpCas9-NG[39], would further expand the range of targetable sequences.

The highly sensitive validation method for off-target mutation, developed in this study, can be combined with genome-wide methods[5,14,15] for highly probable off-target candidate selection. After detecting the most probable off-target candidates, CRISPR amplification can be applied to determine the authenticity of off-target candidates. In addition, the method allows accurate analyses of the tendency of existing gene editors to induce mutations that could not be detected by conventional NGS technologies. Finally, the CRISPR amplification technology can be applied to remove background wild-type DNA and enrich DNA mutations induced by newly developed gene editors, including prime editing[30]. We anticipate that this CRISPR amplification method could be further enhanced by adopting newly developed CRISPR

tools such as specificity increased Cas9[40] or PAM expanded Cas9[38]. We anticipate that the application of the CRISPR amplification method could provide important specificity information to determine whether a given genome-editing approach manifests sufficiently low potential off-target effects for purposes such as usage for gene therapy and ultra-precision genome engineering.

## Methods

**Protein purification.** pET28a–Cas12a (Cpf1), pET28a–SpCas9 and pET28a–eCas9 bacterial expression vectors (target gene was subcloned into pET28a vector from addgene Nos. 79007, 43945, 79145) were transformed into *Escherichia coli* BL21 (DE3) cells for the purification of Cas12a recombinant protein (*Acidaminococcus* sp. (As)Cas12a). The cells were cultured at 37 °C until they reached an optical density of 0.6. After 48 h of isopropyl β-D-thiogalactoside induction, the cultures were centrifuged to remove the medium, and the cells were resuspended in buffer A (20 mM Tris-HCl (pH 8.0), 300 mM NaCl, 10 mM β-mercaptoethanol, 1% Triton X-100, 1 mM phenylmethylsulfonyl fluoride). Then, the cells were disrupted by sonication on ice for 3 min, and cell lysates were harvested by centrifugation at 20,000 × *g* for 10 min. Ni-NTA resin was washed with buffer B (20 mM Tris-HCl (pH 8.0), 300 nM NaCl) and mixed with the cell lysates, and the mixtures were stirred for 1.5 h in a cold room (4 °C). After centrifugation, non-specific binding components were removed by washing with 10 volumes of buffer B (20 mM Tris-HCl (pH 8.0), 300 nM NaCl), and buffer C (20 mM Tris-HCl (pH 8.0), 300 nM NaCl, 200 mM imidazole) was used to elute the Cas12a protein bound to the Ni-NTA resin. The buffer was exchanged for buffer E (200 mM NaCl, 50 mM HEPES (pH 7.5), 1 mM DTT, 40% glycerol) using Centricon filters (Amicon Ultra) and the samples were aliquoted and stored at −80 °C. Purity of the purified protein was confirmed by SDS-PAGE (10%), and protein activity was confirmed by an in vitro PCR amplicon cleavage assay.

**In vitro gRNA synthesis.** For in vitro transcription, DNA oligos containing a crRNA sequence (Supplementary Table 1) corresponding to each target sequence was purchased from Macrogen. crRNA was synthetized by incubating annealed (denaturation at 98 °C for 30 s, primer annealing at 23 °C for 30 s) template DNA was mixed with T7 RNA polymerase(NEB), 50 mM MgCl2, 100 mM rNTP (rATP, rGTP, rUTP, rCTP), 10× RNA polymerase reaction buffer, murine RNase inhibitor, 100 mM DTT, and DEPC at 37 °C for 8 h. Thereafter, the DNA template was completely removed by incubation with DNase at 37 °C for 1 h, and the RNA was purified using a GENECLEAN® Turbo Kit (MP Biomedicals). The purified RNA was concentrated through lyophilization (20,000 × *g*, 1 h, −55 °C, 25 °C).

**Cell culture and transfection.** HEK293FT and U2OS human cells were purchased from the Invitrogen (R70007) and American Type Culture Collection (HTB-96). The cells were maintained in Dulbecco's modified Eagle's medium (DMEM) with 10% FBS (both from Gibco) at 37 °C in the presence of 5% CO2. Cells were subcultured every 48 h to maintain 70% confluency. For target sequence editing, 10^5 HEK293FT or U2OS cells were transfected with plasmids expressing guide RNAs (240 pmol) and genome editors (including AsCas12a (addgene no. 69982), SpCas9 (addgene no. 43945), adenine base editor (addgene no. 112098, ABEmax version), 60 pmol) via electroporation using an Amaxa electroporation kit (V4XC-2032; program: CM-130 for HEK293FT, DN-100 for U2OS). Transfected cells were transferred to a 24-well plate containing DMEM (500 μl/well), pre-incubated at 37 °C in the presence of 5% CO2 for 30 min, and incubated under the same conditions for subculture. After 48 h, genomic DNA was isolated using a DNeasy Blood & Tissue Kit (Qiagen).

**Genotyping by in vitro cleavage.** PCR amplicons were obtained from genomic DNA (from HEK293FT or U2OS) using DNA primers (Supplementary Table 3) corresponding to each target and non-target candidate gene locus. Purified

recombinant Cas12a and crRNA designed to remove DNA other than intracellularly induced mutant DNA were purified and premixed, and incubated in cleavage buffer (NEB3, 10 μl) at 37 °C for 1 h. The reaction was stopped by adding a stop buffer (100 mM EDTA, 1.2% SDS), and DNA cleavage was confirmed by 2% agarose gel electrophoresis. DNA cleavage efficiency was calculated based on the cleaved band image pattern according to the formula: intensity of the cleaved fragment/total sum of the fragment intensity × 100%, using ImageJ software (Java 1.8.0_112).

**Enrichment and validation of Cas12a off-target mutations**. A crRNA corresponding to the nucleotide sequence within the target gene was prepared (Supplementary Table 1), and cells were transfected with plasmids expressing CRISPR–Cas12a and the crRNA to induce genome mutation at the desired site. The target site (RPL32P3 site in human genomic DNA (Supplementary Table 2)) was selected considering the editing efficiency of Cas12a at the given genome location. Cas-OFFinder[18] was used to identify genome-wide candidate off-target sites derived from the target sequence (Supplementary Table 2). To confirm the induction of off-target mutations by Cas12a, genomic DNA was extracted from cells incubated in the presence of plasmids expressing crRNA and Cas12a for 48 h. Intracellular on-target and related candidate off-target genome sites were PCR-amplified (denaturation at 98 °C for 30 s, primer annealing at 58 °C for 30 s, elongation at 72 °C for 30 s, 35 cycles (Supplementary Table 3)). Non-mutated DNA fragments were removed by mixing crRNA (Supplementary Table 1 and Supplementary Fig. 13), Cas12a, and the amplicons, and incubating the mixture in cleavage buffer (NEB3, 10 μl) at 37 °C for 1 h for complete cleavage of wild-type DNA. After this reaction, mutated DNA (targeted by intracellular treatment of Cas12a and not cut by Cas12a effector) was enriched by PCR using nested PCR primers, using only 2% (v/v) of the total reaction mixture. DNA fragments containing mutations (on- and off-target indels) are preferentially amplified over normal DNA. Amplification products were digested and analyzed by 2% agarose gel electrophoresis or NGS with barcode addition using nested PCR (denaturation at 98 °C for 30 s, primer annealing at 58 °C for 30 s, and elongation at 72 °C for 30 s, 35 cycles (Supplementary Table 3)).

**Enrichment and validation of Cas9 off-target mutations**. To use CRISPR–Cas9 to induce mutations, a gRNA was designed and a gRNA expression plasmid was transfected into cells. The target was selected based on a high-editing efficiency of the CRISPR–Cas9 effector (FAT3 site in human genomic DNA (Supplementary Table 2)). Cas-OFFinder[18] was used to identify genome-wide candidate off-target sites (Supplementary Table 2). To confirm off-target mutation by CRISPR–Cas9, genomic DNA was extracted from transfected cells at 48 h after transfection. Intracellular on-target and related off-target candidate genomes site were PCR-amplified (denaturation at 98 °C for 30 s, primer annealing at 58 °C for 30 s, and elongation at 72 °C for 30 s, 35 cycles (Supplementary Table 3)). CRISPR amplification was performed using a crRNA (Supplementary Table 1 and Supplementary Fig. 13) designed to remove DNA other than mutant DNA. Subsequent steps were the same as described for CRISPR–Cas12a off-target mutation analysis.

**Enrichment and validation of ABE off-target mutations**. To use CRISPR–Cas9-based ABE to induce mutations, a gRNA complementary to the target sequence was prepared (Supplementary Table 1). gRNA and ABE expression vectors were simultaneously delivered into cells. The target sequence was selected based on a high-editing efficiency of the ABE, and Cas-OFFinder[18] was used to identify genome-wide candidate off-target sites (Supplementary Table 2). Genomic DNA was extracted from the cells 96 h after transfection. Intracellular on-target and related off-target candidate genomes site were detected by PCR amplification (denaturation at 98 °C for 30 s, primer annealing at 58 °C for 30 s, elongation at 72 °C for 30 s, 35 cycles (Supplementary Table 3)). CRISPR amplification was performed by designing a crRNA (Supplementary Fig. 13 and Supplementary Table 1) to remove DNA other than mutant DNA. Subsequent steps were the same as described for CRISPR–Cas12a off-target mutation analysis.

**Targeted deep sequencing and data analysis**. To confirm whether mutations were introduced in the target genome locus by the CRISPR–Cas12a, CRISPR–Cas9 or adenine base editor, genomic DNA extracted from cells was amplified using DNA primers (Supplementary Table 3) corresponding to target and off-target loci. Nested PCR (denaturation at 98 °C for 30 s, primer annealing at 62 °C for 15 s, and elongation at 72 °C for 15 s, 35 cycles) was performed to conjugate adapter and index sequences to the amplicons. Next, a barcoded amplicon mixture was loaded on mini-SEQ analyzer (Illumina MiniSeq system, SY-420-1001) and targeted deep sequencing was performed according to the manufacturer's protocol. The Fastq data were analyzed with Cas-analyzer (http://www.rgenome.net/cas-analyzer/)[41], and the mutation frequency (%) was calculated (inserted and deleted allele frequency(%)/total allele frequency(%)) and plotted by software (Graphpad prism (8.3.0.538)).

**Reporting summary**. Further information on research design is available in the Nature Research Reporting Summary linked to this article.

## Data availability

All the relevant data support the findings of this study are available from the corresponding author upon reasonable request. All the targeted amplicon sequencing data were deposited at NCBI Sequence Reads Archive database with accession number PRJNA633957 and PRJNA633953. Source data are provided with this paper.

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

## Acknowledgements

We thank Dr. Kyu-Tae Chang for helpful discussions. This research was supported by grants from the National Research Foundation funded by the Korean Ministry of Education, Science and Technology (NRF-2019R1C1C1006603, NRF-2017R1E1A1A01074529, NRF-2018M3A9H3021707, NRF-2019M3A9H110378), the Technology Innovation Program funded by the Ministry of Trade, Industry & Energy (MOTIE, Korea) (20009707), and KRIBB Research Initiative Program (KGM1052021, KGM4252021, KGM5382012).

## Author contributions

Conceptualization, S.-H.K., W.-j.L., and S.H.L.; methodology, S.-H.K. and S.H.L.; software, S.-H.K., J.K.H., and S.H.L.; validation, S.-H.K. and W.L.; formal analysis, S.-H.K. and S.H.L.; investigation, S.-H.K., W.-j.L., J.-H.A., J.-H.L., H.K., Y.O., and S.H.L.; resources, Y.-H.K., Y.-H.P., Y.B.J., and B.-H.J.; data curation, S.-H.K., W.-j.L., and S.H.L.; writing-original draft, S.-H.K. and S.H.L.; writing-review and editing, J.K.H., S.-U.K., and S.H.L.; visualization, S.-H.K., W.-j.L., and S.H.L.; supervision, J.K.H., S.-U.K., and S.H.L.; project administration, J.K.H., S.-U.K., and S.H.L.; funding acquisition, S.-U.K. and S.H.L.

## Competing interests

The authors declare no competing interests.
