## [Peer Review File · Nature Communications]

Reviewers' Comments:

Reviewer #1:

Remarks to the Author:

The authors attempt to apply the CUT-PCR method towards detection of off-target genome editing. By treating the edited genomic DNA with nucleases designed to deplete unmodified, wildtype, sequences and subsequent PCR amplification cycles, even tiny amounts of modified genomic DNA could be detected by high-throughput sequencing.

This work represents an innovative application of an existing technology towards an important field that complements current off-target detection methods. Due to its novelty, ease of use, and extremely high sensitivity, I recommend publication of this work in Nature Communications. However, I feel that the pros and cons of the technology have not been sufficiently described and demonstrated, which reduces the impact of the publication.

Major Concern:

1. As the authors have already alluded to many times, it seems clear and logical that this approach will more efficiently enrich deletions rather than insertions, due to the bias in PCR (smaller fragments are amplified better). In fact, Supplementary Figure 6a shows that insertions that are detected in non-amplified sample are undetectable after amplification. The strengths and weaknesses of this technology will be better illuminated if the authors discuss the effect of insertion mutants and deletion mutants before and after amplification. Perhaps several sites that are known to generate both deletion and insertion mutations can be investigated to ascertain the degree of enrichment and de-enrichment the two types of mutations undergo.

Other concerns:

1. Related to above, the authors state that they observe more enrichment of larger deletion products than deletions containing fewer bases (1-2), and correctly state that mutants with one base deletion may be a substrate during CRISPR enrichment. In fact, it seems that deletions below 1 or 2 bases undergo de-enrichment (Figure 1d, Supplementary Figure 4a,b,c,d,e, Supplementary Figure 5a,b,c, Supplementary Figure 6a). There are some on or off target sites where CRISPR nucleases only generate predominantly 1 base deletion or insertion (Nature 563, 646–651 (2018)) Would the technology fail to enrich (or detect) such sites?

2. The authors rely on the presence of a Cpf1 PAM in the editing window of adenine base editor so that ABE activity abolishes the Cpf1 PAM site. The authors should clearly state that this is a pre-requisite for ABE off-target enrichment. Also, this will only enrich off-target mutants that have base edits at the PAM region of the enrichment guide. The authors should clearly state that mutants that generate adenine editing at other sites in the region will not be enriched. How could one enrich sites that do not have a suitable A/T-rich region that could be used as a Cpf1 PAM?

3. Would the use of high-fidelity versions of CRISPR enzymes reduce the bias towards different indel products during enrichment? More generally, would the choice of CRISPR nuclease during the enrichment affect the result (degree and sensitivity of the enrichment)?

Reviewer #2:

Remarks to the Author:

In this manuscript, Kang et al. developed a CRISPR amplification method to amplify and detect small amounts of off-target mutations induced by CRISPR effectors. Two key factors of the method are accurate in-silico prediction of potential off-target sites and removal of background wild-type DNA. In this study, the authors compared CRISPR amplification with conventional NGS or GUIDE-seq for the detection of off-target mutations and found that extremely small amounts of off-target

mutations below the NGS detection level could be detected by CRISPR amplification, and the sensitivity was augmented with increasing amplification cycles. This study is of potential interest and some suggestions are listed below to revise this study.

Major

- 1) The major finding of this study is that the CRISPR amplification method can be used to detect extremely small amounts of off-target mutations below the NGS detection level. Though the experiments generally support the enrichment properties of this method, the sensitivity and necessity of this method need to be tested at more sites. In this manuscript, off-target mutations at most sites can still be detected by other methods, so the necessity of CRISPR amplification cannot be supported sufficiently.
- 2) The NC data shown in Fig. 2a, 3a and 4a indicated that multiple amplification would induce false positive results and therefore interfere the accuracy of the method.
- 3) The authors suggested that "mutated DNA with larger indels is relatively better amplified", which is inconsistent with the data shown in Fig. S4a-d, S5, S6a. The authors should provide more evidence to support their claim.

Minor

- 1) The editing window of ABE shown in Fig. 4c is confusing. The editing window of ABE has been shown to be positions 4-9 (Gaudelli et al., 2018, Nature, PMID: 29160308) but the authors showed 12-18 in the manuscript. In addition, the authors did not mention which version of ABE they used in their study.

Point by point response to reviewers' comments:

Reviewer #1 (Remarks to the Author):

The authors attempt to apply the CUT-PCR method towards detection of off-target genome editing. By treating the edited genomic DNA with nucleases designed to deplete unmodified, wildtype, sequences and subsequent PCR amplification cycles, even tiny amounts of modified genomic DNA could be detected by high-throughput sequencing.

This work represents an innovative application of an existing technology towards an important field that complements current off-target detection methods. Due to its novelty, ease of use, and extremely high sensitivity, I recommend publication of this work in Nature Communications. However, I feel that the pros and cons of the technology have not been sufficiently described and demonstrated, which reduces the impact of the publication.

Major Concern:

1. As the authors have already alluded to many times, it seems clear and logical that this approach will more efficiently enrich deletions rather than insertions, due to the bias in PCR (smaller fragments are amplified better). In fact, Supplementary Figure 6a shows that insertions that are detected in non-amplified sample are undetectable after amplification. The strengths and weaknesses of this technology will be better illuminated if the authors discuss the effect of insertion mutants and deletion mutants before and after amplification. Perhaps several sites that are known to generate both deletion and insertion mutations can be investigated to ascertain the degree of enrichment and de-enrichment the two types of mutations undergo.

-> We are grateful for the reviewer's suggestion. We conducted additional experiments in order to address major concern 1 and minor concern 1 of the reviewer, concomitantly. Accordingly, we added the related data and explanation in the results and discussion section of the main text (page 9-10, line 212-237; page 14-15, line 340-345) and in (Supplementary Figure 9 – 11).

Discussion line 340-345: "Furthermore, we showed the application of the CRISPR amplification method to the intracellular on- and off-target mutations induced by CRISPR-Cas9. Similar to the findings for CRISPR-Cpf1, mutations below the NGS detection level were amplified to significance level in multiple *in-silico* predicted Cas9 off-target loci. Notably, the CRISPR amplification efficiently enriched ± 1 indel forms in addition to the multiple indel patterns of various sizes generated by CRISPR-Cas9.",

Other concerns:

1. Related to above, the authors state that they observe more enrichment of larger deletion products than deletions containing fewer bases (1-2), and correctly state that mutants with one base deletion may be a substrate during CRISPR enrichment. In fact, it seems that deletions below 1 or 2 bases undergo de-enrichment (Figure 1d, Supplementary Figure 4a,b,c,d,e, Supplementary Figure 5a,b,c, Supplementary Figure 6a). There are some on or off target sites where CRISPR nucleases only generate predominantly 1 base deletion or insertion (Nature 563, 646–651 (2018)) Would the technology fail to enrich (or detect) such sites?

-> We thank the reviewer for the insightful comment. To address the concern, as suggested by the reviewer, we have conducted more amplification experiments for endogenous sites that predominant showed ± 1 indel in the previous study¹. Analyses of the data showed that mutations with single base changes were well amplified even in cases where other mutation types were almost absent. Nonetheless, as the reviewer pointed out, we found that higher ratio of indel-type mutations larger than 1 bp were associated with a tendency to amplify relatively better than DNA containing 1 bp mutations, as the amplification rounds were repeated. We added the related experimental data in main-text (Page 9-10, lane 212-237) and in (Supplementary Figure 9-11).

2. The authors rely on the presence of a Cpf1 PAM in the editing window of adenine base editor so that ABE activity abolishes the Cpf1 PAM site. The authors should clearly state that this is a pre-requisite for ABE off-target enrichment. Also, this will only enrich off-target mutants that have base edits at the PAM region of the enrichment guide. The authors should clearly state that mutants that generate adenine editing at other sites in the region will not be enriched. How could one enrich sites that do not have a suitable A/T-rich region that could be used as a Cpf1 PAM?

We agree with the reviewer's point that the requirement of PAM needs to be clearly stated. Accordingly we added the following in the discussion section of the main text (page 15-16, line 348-366).

"To cover the broad window of adenine mutation, we considered the presence of PAM sequence to optimally design the guide RNAs for the target sequences. Using adequately designed gRNAs and CRISPR amplification, we found that extremely small amounts of off-target base substitution (A>G) below the NGS detection level could be detected, and the sensitivity was augmented with increasing CRISPR amplification cycles. Depending on the target sequences, it may sometimes be difficult to amplify the various types of mutations with Cpf1, which requires TTN or TTTN sequence for recognition of PAM. Nonetheless, there are methods to alleviate the PAM limitations. First, various CRISPR effectors with diverse PAM

sequences could be selectively applied according to target sequences. For example, for the on/off-target DNA enrichment in figure 3, we could apply SpCas9 (NGG recognition as a PAM) to specifically cleave the wild-type DNAs. Second, it is possible to design sgRNAs so that the mutation occurs within the seed sequences proximal to PAM, where CRISPR effectors can discriminate base-pair mismatches with high fidelity. Consistently, CRISPR amplification via designing guide RNAs that perfectly match a wild-type allele could also serve as a method for specific cleavage of wild-type DNA and selective enrichment of mutant allele² via selective cleavage of wild-type DNA. Third, application of engineered SpCas9 with relaxed PAM restriction, such as the recently engineered SpRY³ or SpCas9-NG⁴, would further expand the range of targetable sequences.”

3. Would the use of high-fidelity versions of CRISPR enzymes reduce the bias towards different indel products during enrichment? More generally, would the choice of CRISPR nuclease during the enrichment affect the result (degree and sensitivity of the enrichment)?

We thank the reviewer for the constructive comment. To address the reviewer's point, we conducted experiments to compare the amplification of 1bp mutations between the wild-type SpCas9 and eCas9⁵, a specificity enhanced version of SpCas9 from Feng Zhang's lab. We analyzed and compared the mutant DNA copy number under the same conditions using two versions of SpCas9, and observed that amplification of 1-bp indel by eCas9 was more efficient than wt-SpCas9. Therefore, we basically agree with the reviewer's opinion that the amplification result may change depending on the accuracy of the CRISPR effectors. The results and analyses of the experiments were added in the main text (page 10-11, line 238-250) and supplementary figure 11.

Reviewer #2 (Remarks to the Author):

In this manuscript, Kang et al. developed a CRISPR amplification method to amplify and detect small amounts of off-target mutations induced by CRISPR effectors. Two key factors of the method are accurate in-silico prediction of potential off-target sites and removal of background wild-type DNA. In this study, the authors compared CRISPR amplification with conventional NGS or GUIDE-seq for the detection of off-target mutations and found that extremely small amounts of off-target mutations below the NGS detection level could be detected by CRISPR amplification, and the sensitivity was augmented with increasing amplification cycles. This study is of potential interest and some suggestions are listed below to revise this study.

Major

1) The major finding of this study is that the CRISPR amplification method can be used to detect extremely small amounts of off-target mutations below the NGS detection level. Though the experiments generally support the enrichment properties of this method, the sensitivity and necessity of this method need to be tested at more sites. In this manuscript, off-target mutations at most sites can still be detected by other methods, so the necessity of CRISPR amplification cannot be supported sufficiently.

We are grateful for the reviewer's constructive comment. We sought to address the issue of sensitivity and necessity regarding the off-target validation method through CRISPR amplification. To this end, we conducted additional experiments and analyses of CRISPR amplification on more site of CRISPR-Cas9 based genome editing. Related experimental results were added to the main text (page 8-9. Line 196-211, main text figure 3c, d) and (supplementary figure 7-8).

In the additional experiments, we revisited a previous study that conducted Guide-seq of a target locus (site 4 in the study) in HEK293 cells⁶. The study showed extremely low Guide-seq read numbers for some potential off-target mutations, indicative of very low probability. Conventional deep sequencing method, however, could not confidently verify the presence of the potential off-targets due to their very low allele frequencies in the sequencing data. The limitation was set by the intrinsic detection error of current deep-sequencing technology that affects the fidelity of sequencing read⁷. We performed CRISPR amplification for the off-target sites with off-target read counts that were too low to confirm their authenticity (main text fig. 3c, d). Analyses of the results showed significant amplification for off-target 5 (Guide-seq read count as 3), while no significant amplification was detected for off-target 6 (Guide-seq read count as 3). The results suggested that the CRISPR amplification method could be useful for sensitive discrimination of false-positives from genome-wide off-target candidates.

We also conducted additional amplification experiment and analyzed the indel patterns (Supplementary Fig. 9-11). The analyses showed that some of the indel patterns that existed below the NGS resolution limit (indel frequency <0.5%) were selectively enriched (Supplementary Fig. 11d). The results suggested that the CRISPR amplification could be useful in reliable detection of low-frequency alleles.

We believe that the results of additional experiments showed that CRISPR amplification method is advantageous in verifying potential low-frequency off-targets where the conventional deep sequencing method have been unable to confirm their authenticity.

2) The NC data shown in Fig. 2a, 3a and 4a indicated that multiple amplification would induce false positive results and therefore interfere the accuracy of the method.

We thank the reviewer for the careful comment. According to the reviewer's suggestion we sought to assess the PCR error that could be incorporated by during multiple DNA amplification. To this end, we selected the polymerase carefully to minimize errors during

multiple round of amplifications. In the additional experiments, Phusion DNA polymerase, which is known to generate extremely low rate of mis-incorporation during PCR, from different company (NEB or Thermo) were compared via PCR amplification and analyses of targeted amplicon sequencing.

TERT off4 site

We tested up to third round of CRISPR amplification cycle, and we did not observe any significant amplification for negative controls used throughout this study. All the detected errors were below the NGS resolution limit ($< 0.5\%$). Please see the (supplementary Fig. 12) for the amplification of negative control sites. We also noted that the log scale graph may sometimes overstate the intrinsic background signals that originate from the intrinsic sequencing error (below 0.5 %).

3) The authors suggested that “mutated DNA with larger indels is relatively better amplified”, which is inconsistent with the data shown in Fig. S4a-d, S5, S6a. The authors should provide more evidence to support their claim.

We are grateful to the reviewer for the careful comment. We totally agree with reviewer’s point and changed the original sentence to “The results indicated that a 1-bp alteration from the original sequence within the protospacer region leads to a high probability of re-cleavage by the CRISPR effector, and there was a general tendency for more efficient amplification of large deletions, with some exceptions of preferred sequences (Supplementary Fig. 4a-d, 5, and 6a)”, in main text page 5-6, line 122-126. We also analyzed data of additional CRISPR

amplification experiments on target site where 1bp indel were predominant. We observed relatively increased enrichment of large deletion patterns, compared ± 1 indel patterns, despite the small amount of the DNA (Supplementary Fig. 11d).

Minor

1) The editing window of ABE shown in Fig. 4c is confusing. The editing window of ABE has been shown to be positions 4-9 (Gaudelli et al., 2018, Nature, PMID: 29160308) but the authors showed 12-18 in the manuscript. In addition, the authors did not mention which version of ABE they used in their study.

-> We corrected the sentence in main-text (page 11, line 255-258) and Fig. 4. In addition, we included the version of ABE used in this study in the results and methods section.

References

1. van Overbeek, M. et al. DNA Repair Profiling Reveals Nonrandom Outcomes at Cas9-Mediated Breaks. *Mol Cell* **63**, 633-646 (2016).
2. Kim, J.M., Kim, D., Kim, S. & Kim, J.S. Genotyping with CRISPR-Cas-derived RNA-guided endonucleases. *Nat Commun* **5**, 3157 (2014).
3. Walton, R.T., Christie, K.A., Whittaker, M.N. & Kleinstiver, B.P. Unconstrained genome targeting with near-PAMless engineered CRISPR-Cas9 variants. *Science* **368**, 290-296 (2020).
4. Nishimasu, H. et al. Engineered CRISPR-Cas9 nuclease with expanded targeting space. *Science* **361**, 1259-1262 (2018).
5. Slaymaker, I.M. et al. Rationally engineered Cas9 nucleases with improved specificity. *Science* **351**, 84-88 (2016).
6. Tsai, S.Q. et al. GUIDE-seq enables genome-wide profiling of off-target cleavage by CRISPR-Cas nucleases. *Nat Biotechnol* **33**, 187-197 (2015).
7. Pfeiffer, F. et al. Systematic evaluation of error rates and causes in short samples in next-generation sequencing. *Scientific reports* **8**, 10950 (2018).

Reviewers' Comments:

Reviewer #1:

Remarks to the Author:

The authors successfully addressed all of my concerns, and were able to improve the validity and utility of their claims significantly.

Reviewer #2:

Remarks to the Author:

The authors have improved the study by performing a number of additional experiments and the CRISPR amplification method now makes a useful contribution to off-target mutations detection. Only a few minor points are suggested.

Minor points:

- 1) Line 226-243, "Spacer 8a, 15a" and "Spacer 15A" should be consistent (Supplementary Fig 9-11).
- 2) Please add references for line 73, "in-silico prediction"; line 191-193, "Conventional NGS did not allow detection of the off-target indels with frequencies below the detection limit (indel frequency ~ 0.5)"; line 257-259, "adenine base editor (ABE)".
- 3) Line 190-191, the sentence "and unique indel pattern was detected at off-target site 3 (Fig. 3a, b, Supplementary Fig. 6b)" would be better than the original.

Point by point response to reviewers' comments:

Reviewer #2 (Remarks to the Author):

The authors have improved the study by performing a number of additional experiments and the CRISPR amplification method now makes a useful contribution to off-target mutations detection. Only a few minor points are suggested.

Minor points:

1) Line 226-243, "Spacer 8a, 15a" and "Spacer 15A" should be consistent (Supplementary Fig 9-11).

-> We corrected the sentence in main-text (page 10, line 221-238).

2) Please add references for line 73, "in-silico prediction"; line 191-193, "Conventional NGS did not allow detection of the off-target indels with frequencies below the detection limit (indel frequency ~ 0.5)"; line 257-259, "adenine base editor (ABE)".

-> We added the reference in main-text (page 3, line 70; page 8, line 188; page 11, line 250).

3) Line 190-191, the sentence "and unique indel pattern was detected at off-target site 3 (Fig. 3a, b, Supplementary Fig. 6b)" would be better than the original.

-> We corrected the sentence in main-text (page 8, line 185-186).